# Characterization of Bacterial Transcriptional Regulatory Networks in *Escherichia coli* through Genome-Wide In Vitro Run-Off Transcription/RNA-seq (ROSE)

**DOI:** 10.3390/microorganisms11061388

**Published:** 2023-05-25

**Authors:** Pascal Schmidt, David Brandt, Tobias Busche, Jörn Kalinowski

**Affiliations:** Microbial Genomics and Biotechnology, Center for Biotechnology, Bielefeld University, Universitätsstraße 27, 33615 Bielefeld, Germany; pschmidt@cebitec.uni-bielefeld.de (P.S.); dbrandt@cebitec.uni-bielefeld.de (D.B.); tbusche@cebitec.uni-bielefeld.de (T.B.)

**Keywords:** RNA-seq, run-off in vitro transcription, RNA polymerase, sigma factor, TSS, promoter

## Abstract

The global characterization of transcriptional regulatory networks almost exclusively uses in vivo conditions, thereby providing a snapshot on multiple regulatory interactions at the same time. To complement these approaches, we developed and applied a method for characterizing bacterial promoters genome-wide by in vitro transcription coupled to transcriptome sequencing specific for native 5′-ends of transcripts. This method, called ROSE (run-off transcription/RNA-sequencing), only requires chromosomal DNA, ribonucleotides, RNA polymerase (RNAP) core enzyme, and a specific sigma factor, recognizing the corresponding promoters, which have to be analyzed. ROSE was performed on *E. coli* K-12 MG1655 genomic DNA using *Escherichia coli* RNAP holoenzyme (including σ70) and yielded 3226 transcription start sites, 2167 of which were also identified in in vivo studies, and 598 were new. Many new promoters not yet identified by in vivo experiments might be repressed under the tested conditions. Complementary in vivo experiments with *E. coli* K-12 strain BW25113 and isogenic transcription factor gene knockout mutants of *fis*, *fur*, and *hns* were used to test this hypothesis. Comparative transcriptome analysis demonstrated that ROSE could identify *bona fide* promoters that were apparently repressed in vivo. In this sense, ROSE is well-suited as a bottom-up approach for characterizing transcriptional networks in bacteria and ideally complementary to top-down in vivo transcriptome studies.

## 1. Introduction

Most bacteria must quickly adapt to changing environmental conditions, such as temperature, pH, osmolarity, or nutrient availability. Gene expression can be regulated at every step, from transcription to post-translational processing; however, transcription and, more accurately, transcription initiation is the first and most widely regulated step in bacteria [1]. One important mechanism of regulating transcription initiation is either repression or activation by DNA-binding transcription factors (TFs). Another widely found mechanism of transcription regulation is using multiple forms of RNA polymerase consisting of RNA polymerase core enzymes and different sigma factors, each allowing specific recognition of distinct promoter sequences. Bacteria can quickly reprogram their transcriptional landscape using several sigma factors to cope with extracellular and intracellular stress factors and the changing environment [2].

In *E. coli*, seven sigma factors have been described: the primary sigma factor σ^70^ and six alternative sigma factors (σ^54^, σ^38^, σ^32^, σ^28^, σ^24^, and σ^19^). σ^70^ is the housekeeping sigma factor responsible for the transcription of most growth-related genes. The consensus sequence of σ^70^-dependent promoters (5′-TTGACA-17-TATAAT) is well known and highly conserved among σ^70^-dependent promoters. However, the −10 region, which occurs about seven base pairs upstream of the transcription start site, shows a higher level of conservation than the −35 region in *E. coli* [3]. 

The run-off transcription-microarray analysis (ROMA) method has been described for genome-wide analysis of transcription regulated by sigma factors [4]. Here, purified RNA polymerase holoenzyme is used for the in vitro transcription on fragmented genomic DNA. Transcribed mRNA species are then identified by DNA microarray hybridization, representing all genes of the respective genome. After the initial ROMA experiments conducted for the transcriptional profiling of *Bacillus subtilis*, the ROMA method was successfully applied to *E. coli* and was used to disentangle the overlapping σ^70^ and σ^38^ regulons [5]. ROMA allows for the investigation of the direct effects of different sigma factors without regulatory proteins. 

ROMA also has limitations, including the lack of single-nucleotide resolution and transcriptional read-through at convergently oriented genes, possibly leading to false-positive signals. MacLellan et al. observed the extended transcription of up to 10 kb downstream of the active promoters [6]. Although co-transcription of convergent genes regularly occurs in vivo, the frequency is significantly higher in vitro. Therefore, specifically activated promoters identified by ROMA must be confirmed by alternative methods, such as in vivo reporter fusions or single-promoter in vitro transcription [4]. 

We developed run-off transcription/RNA-seq (ROSE) to overcome these limitations. ROSE employs genome-wide in vitro transcription with isolated RNA polymerase and genomic DNA. In vitro transcribed RNA analysis includes the preparation of native 5′-end-specific transcript libraries [7] and subsequent transcriptome sequencing. Mapping the sequenced 5′-ends to the genome provides distinct read stacks at the transcription start site of a given mRNA, which enables the detection of promoter sequences with single-nucleotide resolution. The method was initially developed in the frame of a PhD thesis in 2013 by combining the *E. coli* core RNA polymerase with the ECF sigma factors of *Corynebacterium glutamicum* [8]. A similar technique, regulon identification by in vitro transcription-sequencing (RIViT-seq), was recently described [9]. RIViT-seq was used to identify new target genes of 11 different sigma factors in *Streptomyces coelicolor*. Since there are several technical differences between ROSE and RIViT-seq, especially with the preparation of the primary transcript libraries and the determination of transcription start sites (TSSs), we like to keep the name ROSE and refer to the technical differences compared to RIViT-seq and their implications in more detail below. ROSE and RIVit-seq can be regarded as different flavors of a ‘bottom-up’ approach to transcription regulatory networks in bacteria. Therefore, they are ideally complementary to ‘top-down’ transcriptome analysis in vivo, e.g., by comparing wildtype and transcription factor mutant strains.

In this study, we demonstrate the power of ROSE by using the native *E. coli* RNA polymerase holoenzyme with σ^70^ to characterize bacterial transcriptional regulatory networks of *E. coli*. 

## 2. Materials and Methods

### 2.1. In Vitro Transcription on Genomic DNA Fragments

Genomic DNA from *E. coli* K-12 MG 1655, cultivated overnight on solid lysogeny broth medium [10], was isolated with three different approaches: the Quick-DNA Universal Kit (Zymo Research, Irvine, CA, USA), the NucleoSpin Microbial DNA Kit (Macherey-Nagel, Düren, Germany) and a phenol–chloroform isoamyl alcohol DNA extraction. Each isolation method was used twice to get three biological and two technical replicates for ROSE. For the phenol–chloroform extraction, the cells were treated with lysozyme, RNAse H, and proteinase K to open the cells before the phenol extraction of the DNA. All three isolation methods were made with two technical replicates. DNA was fragmented randomly to an average size of 6 kb using gTubes (Covaris, Woburn, MA, USA). Size distribution of the fragmented template DNA was checked using an Agilent Bioanalyzer with the high-sensitivity DNA kit. 1 µg of template DNA was used for a single in vitro run-off transcription reaction. In vitro run-off transcription was performed in an *E. coli* RNA polymerase buffer (New England Biolabs, Ipswich, MA, USA). Template DNA and reconstituted RNAP holoenzyme were incubated at 37 °C for 15 min, followed by the addition of NTPs to a final concentration of 200 nM each to start the transcription reaction. After 60 min, in vitro run-off transcription was terminated by five-minute incubation at 65 °C. Template DNA was digested with DNase I (Roche, Basel, Switzerland) immediately after in vitro transcription (30 min, 25 °C). In vitro transcribed RNA was purified using a Qiagen RNeasy MinElute Kit, including a second DNase digestion, and eluted in nuclease-free water. At least three independent in vitro transcription reactions were combined to obtain sufficient RNA for transcriptome sequencing. Nucleic acid concentration and purity were determined with an Xpose spectrophotometer. RNA quality and size distribution were checked on an Agilent Bioanalyzer 2100 using the RNA Pico assay. PCR amplification has ruled out residual DNA contamination with specific primers binding to the *E. coli* genome. 

### 2.2. Cultivation of the E. coli Knockout Strains

The following transcription factor deletion mutants were derived from Coli Genetics Stock Center (CGSC): #7636 (BW25113), #8758 (Δ*fur*), #10443 (Δ*fis*), and #9111 (Δ*hns*). Deletion strains were cultivated in liquid LB broth containing 50 mg/mL kanamycin in shaking flasks (37 °C, 180 rpm). After inoculation to an OD_600_ of 0.05 from an overnight liquid culture, 2 mL cell suspensions were harvested at an OD_600_ of 0.8 in the exponential growth phase and immediately frozen in liquid nitrogen and stored at −80 °C. For the wildtype strain, another sample was taken after cells reached the stationary growth phase.

### 2.3. Construction of Primary Transcript Libraries

A previously established procedure has been used to prepare the mRNA for sequencing and to enrich primary, unprocessed transcripts [7]. Shortly, 100 ng purified in vitro-transcribed RNA was fragmented to an average size of 500 nt. For in vivo-based libraries, 5 µg total RNA was initially subjected to Ribo-Zero treatment (Illumina, San Diego, CA, USA) to deplete rRNA before fragmentation. Next, transcripts harboring a 5′ di- or monophosphate were digested by terminator exonuclease (Epicentre). RNA index adapters were ligated for noise reduction in the sequencing. Furthermore, the indexed transcripts were treated with RNA 5′-polyphosphatase to enable the ligation of specific sequencing adapters. RNA adapters were ligated, and RNA was then reverse-transcribed to cDNA using a sequence-independent loop adapter. With 18 cycles of PCR, cDNA was amplified using barcoded primers to generate a multiplexed cDNA library ready for sequencing with Illumina technology.

### 2.4. Sequencing and Data Processing

High throughput single-end sequencing (1 × 75 bp) was conducted on Illumina MiSeq (San Diego, CA, USA). Reads were quality trimmed using Trimmomatic v0.3.5 [11] with the following parameters: TRAILING:3 and MINLEN:39. Forward reads were mapped to the respective *Escherichia coli* K-12 MG1655 (U00096.3) genome sequence using bowtie2 v2.3.0 [12] in single-end mode, as the 5′-ends of transcripts were of particular interest for TSS identification. 

### 2.5. Sequence Analysis

After identification of transcription start site (TSS) positions with ReadXplorer v2.2.3 [13], upstream sequences (−1 to −49) were subjected to motif enrichment analysis using either Improbizer: [14] https://users.soe.ucsc.edu/~kent/improbizer/improbizer.html (accessed on 24 May 2023) or MEME v4.1.2 [15]. Finally, sequences were aligned at conserved motif positions, and sequence logos were generated using WebLogo v3.7.0 [16]. For higher accuracy, only TSS positions detected in at least four of the six ROSE approaches were considered for sequence analysis.

## 3. Results

### 3.1. Development of ROSE and Application to the Analysis of σ^70^-Dependent Promoters in E. coli

The ROSE method was developed based on ROMA [4], which used DNA microarrays for genome-wide run-off transcription analysis. Accordingly, run-off transcription assays were performed employing the commercially available σ^70^-saturated form of *E. coli* RNA polymerase (Eσ^70^). In contrast to ROMA and RIViT-seq [9], template DNA has been fragmented to an average size of 6 kb by shearing instead of restriction enzyme treatment to avoid bias by unequal distribution of restriction enzyme recognition sites or by cutting within a promoter. In addition, commercially available Tris HCl buffer (NEB, Ipswich, MA, USA) was used instead of the potassium glutamate-based buffer system previously used in ROMA. 

Although ROSE and RIViT-seq aim for a genome-wide in vitro-transcriptome, the two approaches have distinct technical differences (see Appendix A for a complete list). The focus of ROSE is the construction of high-quality primary transcript libraries. Therefore, the digestion of RNA having a 5′ di- or monophosphate is necessary to maximize the purification of unprocessed, primary bacterial transcripts. Moreover, ROSE uses index adapter ligation to reduce the noise in the sequencing, and the TSSs were identified in an automated fashion using the software ReadXplorer v2.2.3 [13,17]. 

Before sequencing, in vitro transcribed mRNA was subjected to native 5′-end-specific transcript library preparation [7]. Sequencing on Illumina MiSeq yielded around 2 million reads per library (see Appendix A), which were quality-filtered and mapped to the respective reference genome (U00096.3). Three different approaches were tested for isolating *E. coli* K-12 MG1655 chromosomal DNA. The different isolation methods did not result in notable differences in the quality or the distribution of reads (see Appendix A). 

Mapped reads were visualized using ReadXplorer v2.2.3 software [17], and transcription start site (TSS) detection was performed with the same tool and automatic parameter estimation (see Appendix A). The automatic TSS detection identified 3226 possible TSSs in at least four of the six ROSE runs. Depending on their location relative to known genes, the TSSs were classified into four categories according to Sharma et al. [18]: primary TSSs (44.6%), intragenic TSSs (24.4%), antisense TSSs (27.1%) and orphan TSSs (3.9%). Primary TSSs comprise all TSSs located in a suitable distance and direction to a protein-coding region or a known transcript. Intragenic TSSs are located within a coding sequence. In sense orientation, antisense TSSs are situated on the opposite strand of a protein-coding region up to 100 bases upstream or downstream (±100 nt), and orphan TSSs do not meet any of these criteria. 

To validate the suitability of the ROSE method for promoter identification, upstream sequences of 50 nt lengths (positions −1 through −49 relative to the TSS) were extracted for further analysis. All 3226 putative promoter sequences were subjected to motif enrichment analysis using Improbizer [14]. Two distinct motifs corresponding to −10 and −35 regions of σ^70^-dependent promoters were detected independently (Figure 1). As expected from previous studies, the −10 region shows a considerably higher level of conservation [19]. In total, 3128 putative promoter sequences contained a region similar to the σ^70^ −10 consensus. 

In contrast, a −35 consensus motif, namely a conserved ttGA about 35 nucleotides upstream of the transcription start site, was derived from 2922 promoter sequences. A total of 2838 putative promoter sequences contained both −10 and −35 regions. Only 11 sequences did not resemble either the −10 or the −35 consensus sequence. 

It is apparent that Eσ^70^ recognizes natural σ^70^-dependent promoters in vitro with high specificity and initiates transcription at well-defined nucleotides. Transcription initiation occurred preferentially at purine bases (A/G), which was observed in 81.0% (50.3% A and 30.7% G) of the detected promoters. Interestingly, the base directly upstream of the TSS at position −1 prefers pyrimidine bases, with 77.3% of the promoters harboring T (41.3%) or C nucleotides (36.0%) at the respective position. Both findings align with in vivo transcriptional profiling studies, reporting similar nucleotide preferences of 78.6% purine bases at +1 and 80.2% pyrimidine bases at −1 [19,20].

### 3.2. Detailed Promoter Analysis by Comparison to Experimentally Characterized Promoters Listed in RegulonDB

The genome of *Escherichia coli* K-12 MG1655 contains 4146 genes organized in 2376 transcription units. In total, 1523 transcription units are monocistronic, whereas 853 operons have more than one gene [21]. Thus, at least 2376 primary TSSs are expected to be found, except for, possibly, the TSSs of promoters that need to be activated by factors not contained in the in vitro transcription assay. The RegulonDB database [22] includes the most comprehensive information regarding the transcriptional regulation of *E. coli*, including experimentally determined transcriptional start sites of the strain K-12 MG1655. In a subset of the database, TSSs are assigned to the different sigma factors and provided with a level of evidence (confirmed, strong, or weak), depending on the informative value of the method for TSS identification. For the following comparison, only those TSSs were considered to belong to the classes “Confirmed” or “Strong”. In addition, to cope with different experimental methods of TSS identification and issues of the template, such as the degree of supercoiling, a deviation of three nucleotides in either the upstream or downstream direction has been allowed to compare two TSS positions. The mapped TSSs show a clear peak, with 64.7% having zero and 7.0% having one nucleotide deviation in either direction (See Appendix A). 

In RegulonDB, 881 TSS are classified as σ^70^-dependent, and 352 (40.0%) were also identified in the ROSE-Eσ^70^ experiment. A total of 30 TSSs found in our ROSE experiment are assigned to other sigma factors in the database with no affiliation to σ^70^. In total, 25 of these TSSs are classified as σ^38^-, and the other five as σ^32^-dependent promoters; however, it is known that the consensus sequence of σ^38^-dependent promoters is similar to the σ^70^ consensus sequence, and a clear distinction between both promoter sets cannot be made [23]. Therefore, promoter sequences identified by ROSE-Eσ^70^ but listed as σ^38^-dependent in RegulonDB were compared to those of σ^38^-dependent promoters that ROSE did not recognize. Again, 50 nt upstream of the TSSs have been extracted and analyzed for conserved motifs. Comparing the resulting motifs clearly shows differences, mainly in the −10 regions. The presumed σ^38^-dependent promoters show conserved bases at all positions from −12 through −7 (TATACT), whereas in the σ^38^-dependent promoters not detected by ROSE-Eσ^70^, only the bases at −12, −11, and −7 are conserved (TANNNT). Additionally, there is a C at position −13, upstream of the −10 region, described earlier as a distinct sequence characteristic in σ^38^-dependent promoters [24]. Another distinguishable feature of the exclusively σ^38^-dependent promoters is a highly conserved GC at positions −33/−32 (ttGC), occurring in most σ^38^-dependent promoter sequences, with a higher conservation of the TT at position −35/−34 in the promoters present in ROSE-Eσ^70^ (Figure 2).

Following the same reasoning, five predicted false-positive σ^32^-dependent promoters were compared to 66 σ^32^-dependent promoters from Regulon DB that ROSE did not detect. Due to the low number of five false-positive promoters, no precise consensus sequence could be identified in the −10-region; however, the similarity of the −10 region of these false-positive σ^38^-dependent promoters to those of σ^70^-dependent promoters suggests that ROSE-Eσ^70^ falsely identifies these promoters as σ^70^ promoters, possibly due to in vivo regulatory mechanisms in the in vitro ROSE-Eσ^70^ system. 

### 3.3. Comparison of the ROSE Data to Existing Comprehensive Genome-wide In Vivo RNA-seq Data Sets of E. coli K-12 MG1655

To date, genome-wide transcription start site determination is mainly performed by analyzing in vivo transcribed mRNA via approaches, such as dRNA-seq [18,25]. To assess the sensitivity and selectivity of ROSE, results were compared to a transcriptome study by Thomason et al. [20] and another high-throughput transcription initiation mapping study included in RegulonDB [22]. Both studies were conducted on shaking flask cultivations of *Escherichia coli* MG1655 in different media. After enriching 5′-triphosphorylated RNA species and high-throughput sequencing, they detected 14,865 TSSs and 5197 TSSs, respectively [20,22]. Although both studies relied on transcriptome sequencing for TSS identification, their suitability for validating ROSE is limited because no specific sigma factor-promoter interaction can be examined. However, as we performed ROSE using the primary sigma factor σ^70^, it was assumed that there was reasonable overlap in detected TSSs.

Comparing the three TSS datasets showed that 2006 (62.2%) of the TSSs detected by ROSE were also determined by Thomason et al., while 168 further TSSs are confirmed by the study included in RegulonDB. A set of 755 TSS was contained in all three datasets (Figure 3). Again, a deviation of three nucleotides has been allowed in the comparison of two TSS positions. Here, ROSE-Eσ^70^ and RegulonDB exactly matched in 76.0% (±1 bp: 13.9%) of the overlapping TSSs, while ROSE-Eσ^70^ and Thomason et al. had an exact match at 86.2% (±1 bp: 7.6%) of the TSSs.

### 3.4. Transcription Start Sites of Promoters That Are Repressed under Standard In Vivo Assay Conditions Are Comprehensively Identified in ROSE Experiments

By design, ROSE should be able to identify two classes of promoters not represented in RegulonDB. The first class comprises those present in the *E. coli* genome but not described in existing TSS mapping studies. The second class includes those that are repressed or not activated under standard in vivo testing conditions. In total, 2303 transcriptional start sites detected by ROSE-Eσ^70^ are yet undescribed, according to RegulonDB. Thomason et al. identified 1254 of those TSSs in vivo. The remaining 1049 upstream regions were subjected to motif enrichment analysis using Improbizer [14]. To remove possible background signals, the sequences have been sorted by the −10-region score given by Improbizer, which corresponds to the similarity of a given sequence to the detected consensus motif. A randomized control run yielded a 95% confidence score of 6.20 for a given sequence. After filtering with this value as a cut-off, 598 sequences remained containing a precise σ^70^ consensus sequence. Due to this, it can be speculated that these promoters were repressed under the conditions tested in the in vivo studies. Manual inspection showed regulator binding sites around many of these promoters, suggesting that transcription from those promoters is prevented in vivo by known transcriptional regulators, such as H-NS, Fur, or Fis. We performed in vivo experiments for each of the three regulators with defined transcription factor knockout mutants from the KEIO collection [26] to validate the results observed with ROSE. The knockout mutants were JW1225-2 for Δ*hns*, JW0669 for Δ*fur*, and JW3229-1 for Δ*fis*. Sequencing on Illumina MiSeq yielded, on average, 0.91 million reads per library (See Appendix A). The mapped reads were visualized using the ReadXplorer v2.2.3 software [17], and transcription start site (TSS) detection was performed with the same tool and automatic parameter estimation (see Appendix A). In the in vivo transcriptome analysis, multiple TSSs were identified for all three transcriptional regulators with clear TSS signals present in the knockout mutant strains and absent in the wildtype in vivo. These signals are also identified by ROSE-Eσ^70^ proving its ability to detect TSSs repressed in vivo. Most identifications confirm known regulatory sites documented in RegulonDB [22] (Figure 4). In addition, we found four potential genes repressed for Fur, ten potentially repressed genes for Fis, and nine potentially repressed genes for H-NS (also see Appendix A for details). We will now describe two promoter regions for each transcriptional regulator, H-NS, Fur, and Fis.

The genes *stpA* (b2669) and *ftnA* (b1905) are both negatively regulated by H-NS, a global transcriptional silencer [28], which is involved in the regulation of 5% of all *E. coli* genes [29]. In both cases, H-NS binds upstream of the TSS and leads to a repression of transcription [30,31] (Figure 4A). The gene *stpA* has a TSS at position 2,798,556 and a perfect σ^70^-like −10 region (TATAAT). The gene *ftnA* has a TSS at position 1,988,682 and has a complete σ^70^-like promoter (TTGCAA-16-TATAGT). Both genes showed no transcription in the wildtype strain, but transcriptional activity was measured in the Δ*hns* knockout strain and in the ROSE approach (see Appendix A). Moreover, both genes were also described by Thomason et al. and RegulonDB.

The TSS of *yjjZ* (b4567) has already been described for genomic position 4,605,777 in the *E. coli* MG1655 genome. According to *EcoCyc*, there are two ferric uptake regulator (Fur) binding sites in the vicinity of the transcription start site of *yjjZ* [27,32] (Figure 4B). Although the respective promoter harbors a σ^70^-like consensus sequence (TTGCAA-18-TATGAT), Thomason et al. did not detect a transcription start site for *yjjZ*, suggesting efficient transcriptional repression in vivo. This has been validated in our in vivo experiment, where the wildtype strain has shown no activity of the *yjjZ* gene; however, in the Δ*fur* knockout strain, transcription from the σ^70^ promoter is measurable. Moreover, the TSS has been identified clearly in vitro (1494 read starts) using the ROSE method. (see Appendix A). Another example of a gene activated by the regulator Fur is the gene *fepA* (b0584) [33,34] (Figure 4B). The *fepA* promoter has a σ^70^-like consensus sequence (TTGCAG-14-TATTAT) and was not detectable in vivo in the wildtype strain; however, both the Δ*fur* knockout strain (508 read starts, in vivo) and ROSE (348 read starts, in vitro) show transcriptional activity for the gene *fepA* (see Appendix A).

The gene for the DNA-binding transcriptional dual regulator GlcC has a TSS at position 3,128,206. It has an unusual −10 region (CATAAT) and a −35 sequence (TTAACT). As stated in *EcoCyc*, the gene’s promoter region has four binding sites for the global regulator Fis [35] (Figure 4C), which causes gene repression. This repression has been validated in the in vivo experiment using the wildtype strain and the Δ*fis* knockout mutant. The wildtype strain showed minimal read starts (7 read starts) for the gene in vivo. In the knockout strain, the amount of read starts was five times higher than in the wildtype strain, demonstrating the higher transcription of *glcC* in the absence of the Fis regulator; however, the most read starts and the strongest transcription of the gene were identified by the in vitro ROSE approach (493 read starts) (see Appendix A). Another exciting gene is *aer* (b3072), which shows a clear transcription start site in ROSE, the Δ*fis* knockout strain at position 3,219,346, and that it is harboring a σ^70^-like consensus sequence (TTGTGC-19-TAACAT). This transcription start site is also described in the publication of Thomason et al. but is not defined in RegulonDB. Nevertheless, RegulonDB contains a Fis binding site with an unknown function upstream of the *aer* gene (Figure 4C). ROSE and the Δ*fis* knockout strain showed transcriptional activity, but there was no transcription in the wildtype strain, suggesting that Fis is a transcriptional repressor of *aer* (see Appendix A). 

The gene *ndh* of *E. coli* expresses the NADH dehydrogenase II. The corresponding promoter P*ndh* is located at position 1,165,992 of the genome and is harboring a standard σ^70^-like consensus sequence (TTGGTA-21-TATTCT). This gene is negatively regulated by multiple transcription factors, such as FNR [36], Fur-Fe^2+^ [37], and NsrR [38]. Due to the high number of different repressors of *ndh*, no transcription was detectable in the *E. coli* wildtype strain or the single knockout strains in vivo. However, the ROSE method showed a distinct TSS at the known position of P*ndh* with over 200 read starts (Figure 5).

These findings underline that the bottom-up approach employed within ROSE aids the identification of previously undetected TSSs, especially those that are repressed or not activated under a given in vivo testing condition.

### 3.5. Promoters Activated by Transcriptional Regulators In Vivo Are Not Identified In Vitro

A different type of σ^70^-dependent promoter comprises those specifically activated by transcriptional regulators in vivo, possibly allowing for lesser conservation of promoter motifs. For example, the well-known promoter of the *araBAD* operon (CTGACG-18-TACTGT) of *E. coli* can be activated and repressed by the transcriptional regulator AraC in vivo, depending on the availability of arabinose [39,40]. It is furthermore activated by the cAMP receptor protein (CRP) in vivo [41,42,43]. Since none of these regulators are included in the ROSE in vitro transcription assay, neither activation nor repression of pBAD should occur. Interestingly, no TSS has been identified upstream of the *araBAD* operon by ROSE-Eσ^70^, suggesting that CRP and/or AraC activation is critical for transcription initiation at pBAD. Another instance is the σ^70^-dependent promoter of *csiE* (b2535), known to be activated by both CRP and H-NS in vivo [44,45]. Dual activation allows for relatively weak −10 and −35 hexamers (TTCCCT-18-AACTTT). Consequently, the respective TSS at position 2,665,401 is included in both in vivo-based studies but was not detected by ROSE-Eσ^70^. The σ^70^-dependent promoter of *alkA* is activated upon binding to Ada, a DNA repair protein, which is a critical component of the adaptive response [46,47]. The promoter of its TSS at position 2,147,559 harbors a well-conserved −10 region (TATGCT) but has no −35 region. In contrast to both in vivo studies, it is not detected by ROSE-Eσ^70^, obviously requiring activation by Ada. In conclusion, ROSE robustly and comprehensively identifies *bona fide* promoters and those potentially repressed under in vivo conditions. It also allows drawing conclusions from negative results, predicting efficient activation in vivo.

## 4. Discussion

In this study, we developed the ROSE method for genome-wide in vitro transcriptional profiling and validated it by exploring the σ^70^ regulon of *E. coli* K-12 MG1655.

Like ROMA [4], ROSE is a bottom-up approach aiming to assemble the transcriptional machinery from a few simple parts. It perfectly complements in vivo transcriptome profiling, which can be regarded as a top-down approach. The latter represents the much more complex situation that includes indirect interactions, making such data harder to interpret.

In vitro transcription analyzed by genome-wide methods, as in ROMA [4], RIViT-seq [9], or ROSE, provides several benefits compared to traditional single gene-oriented approaches. The in vitro methods are free from transcriptional repression, allowing for the detection of promoters negatively regulated at standard cultivation conditions. These simple bottom-up approaches enable the precise dissection of overlapping sigma factor networks by employing single sigma factor proteins in the assay, thereby focusing the observation on the direct effects of the respective regulators. It has furthermore been shown by the pioneering work of MacLellan et al. [4] that linear DNA conformation and relatively low complexity of in vitro systems maintain the specificity of transcription initiation. In contrast to ROMA, the particular RNA-seq protocol used here provides clear evidence that even the transcriptional start nucleotide is the same as in vivo. Furthermore, single-nucleotide resolution allows direct TSS identification and, consequently, the derivation of promoter sequences and their consensus motifs. Technically, ROSE has some additional features to the RIViT-seq technique. First is the shearing of the DNA, avoiding bias by restriction enzyme digestion. Second is the focus on establishing an enriched unprocessed primary transcript library by removing transcripts with 5′ di- and monophosphate ends. With the usage of the index adapter before the Illumina adapter ligation, some noise in the sequencing is reduced, leading to a higher quality of the sequenced library. Whereas ROSE is optimized for high-accuracy TSS detection, RIViT-seq has an advantage in differential expression analysis by using whole transcriptomics as an additional data set to its primary transcript libraries. As such, it might also detect 5′-ends of transcripts prone to swift 5′-end decay. Moreover, ROSE and RIViT-seq have individual approaches to identifying transcription start sites. Therefore, combining both techniques could result in a more comprehensive determination of novel TSSs and the complete identification of target genes of interesting transcription factors.

The ROSE analysis of the *E. coli* σ^70^ regulatory network proved consistent with in vivo-based transcriptome studies. Accordingly, 2174 of 3226 TSSs (67.4%) identified by ROSE were also described earlier in comprehensive reference studies [20,22], while ROSE-Eσ70 additionally identified 598 promoters with conserved σ70 motifs. One major cause for differences is likely the simple composition of the ROSE bottom-up in vitro transcription assay, which does not resemble the complex in vivo situation by design. Nonetheless, genome-wide in vitro transcription using homologous *E. coli* RNAP showed high specificity with only 11 detected TSS lacking a typical σ^70^ promoter motif. Interestingly, ROSE-Eσ^70^ data also contained TSS earlier assigned to other sigma factors (σ^38^, σ^32^). Apart from possible dual recognition in vivo, linear template DNA confirmation could have facilitated this issue, as the σ^70^-containing holoenzyme is known to preferentially initiate transcription on more highly supercoiled DNA [48,49]; however, as proposed earlier and confirmed by recent studies, actively transcribing RNA polymerase produces a (+) supercoiling domain ahead and a (-) supercoiling domain behind it, even on linear template DNA [50,51,52]. This activates supercoiling-dependent promoters like the *leu-500* promoter from *E.* coli [52] and suggests that the linear template within the ROSE assay exhibits a certain degree of supercoiling and, therefore, supercoiling-dependent promoters should be, in principle, identified in ROSE experiments. Furthermore, complementary in vivo experiments were used to demonstrate the identification of promoters, which are repressed under standard testing conditions in vivo with ROSE. In vivo knockout strains showed no expression of the respective knockout genes, indicating the knockout’s functionality. The in vivo experiments demonstrated that the three tested transcription factors, Fur, Fis, and H-NS, lead to a repression of specific genes, which could only be identified with the transcription factor knockout strain in vivo. Nevertheless, we demonstrated that ROSE enables the identification of TSS detectable in vivo only in specific knockout strains. However, it should be noted that some genes activated by these factors were not detected by ROSE, as were genes requiring other sigma factors such as σ^24^ or σ^38^ for transcription initiation. Despite this, ROSE-σ^70^ was able to identify several novel potentially regulated promoter sites for all three tested transcription factors. Interestingly, our research also revealed the presence of genes, like *yjjZ*, which were believed to be regulated by one of the tested factors but did not exhibit any read starts in our in vivo experiments. Nevertheless, these genes were shown to have noticeable TSSs in ROSE. These are more complex promoters with multiple repressor binding sites, and numerous knockouts would be needed to identify this TSS in vivo. For example, the promoter P*ndh* of the gene *ndh* is repressed by different regulators like FNR [36], Fur-Fe^2+^ [37], and NsrR [38] and showed no activity in all strains in vivo. However, ROSE identified a TSS for the *ndh* gene and demonstrated the method’s power due to its minimalistic construction (Figure 5), however, the minimalistic structure of the system leads to limitations regarding more complex regulatory systems. This results in a lack of identification of promoters that need activators, like the promoter of the *araBAD* operon or the promoter of the gene *csiE*.

Therefore, expanding ROSE appears possible by adding regulators, such as transcription factors or metabolite effectors, to directly investigate their influence on transcriptional regulation. This has, for instance, been demonstrated for the regulator protein DksA [53] and the small alarmone ppGpp [54,55] in single-promoter in vitro transcription assays. Moreover, it became apparent that genome-wide in vitro transcription studies are not limited to *E. coli* genomes. For example, the *E. coli* RNAP holoenzyme has also been used successfully for in vitro transcription of promoters from other bacteria [56,57]. Furthermore, RIViT-seq [9] demonstrated that a reconstitution of the *E. coli* RNAP core enzyme with sigma factors of *Streptomyces coelicolor* is possible for genome-wide in vitro transcription studies. However, problems might arise if the interaction of the organism-specific RNAP core enzyme with distinct promoter motifs or sigma factors is crucial for transcription. Therefore, homologous RNAP complexes have been isolated and functionally tested for a broad spectrum of bacteria like *Bacillus subtilis* [58], *Pseudomonas aeruginosa* [59], *Mycobacterium tuberculosis* [60] or *Corynebacterium glutamicum* [61] and can be used in in vitro transcription systems. Therefore, ROSE and RIViT-seq could be applied to almost any other bacteria, including those with highly complex sigma factor networks, bacteria without developed genetic engineering technologies, or highly pathogenic ones.

## 5. Conclusions

The global in vitro transcription method ROSE presented in this study is the perfect addition to classical global in vivo and local in vitro transcription assays due to its simplicity and wide range of possible applications. It can be used to identify the primary effects of different sigma factors and their binding motifs with single-nucleotide resolution. We are expanding the technology by transferring it to other bacteria and adding regulatory proteins and small molecules.

## Figures and Tables

**Figure 1 microorganisms-11-01388-f001:**
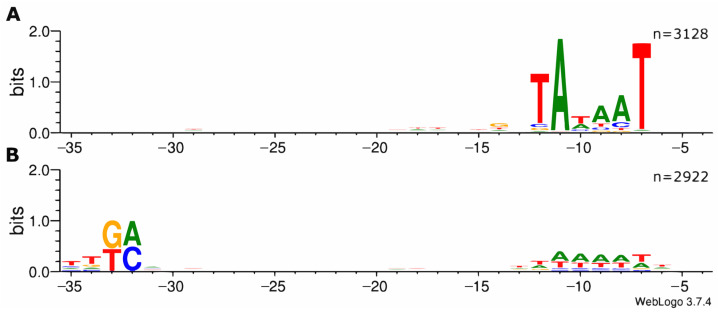
Distribution of nucleotides within the −35 and −10 core regions of *Escherichia coli* σ70-dependent promoters detected via ROSE-Eσ70. Upstream sequences of 3226 TSS (−1 to −49 nt) have been analyzed for enriched sequence motifs using Improbizer [14]. Sequence logos were derived using WebLogo v3.7.4 [16]. (**A**) In total, 3128 putative promoter sequences were aligned at conserved −10 regions. (**B**) In total, 2922 putative promoter sequences were aligned at conserved −35 regions.

**Figure 2 microorganisms-11-01388-f002:**
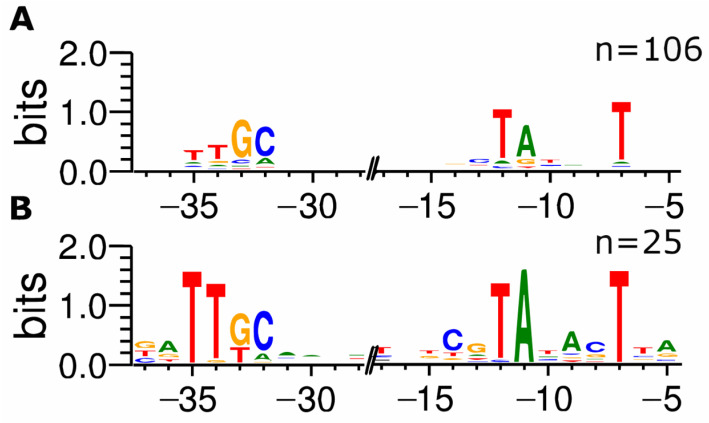
Distribution of nucleotides within putative σ38-dependent *Escherichia coli* promoters detected via ROSE-Eσ70. Putative σ38-dependent promoters have been extracted from RegulonDB [22], and promoter motifs upstream of 106 TSS that were absent (**A**) and 25 TSS that were present (**B**) in the ROSE-Eσ70 dataset were visualized using WebLogo v3.7.4 [16].

**Figure 3 microorganisms-11-01388-f003:**
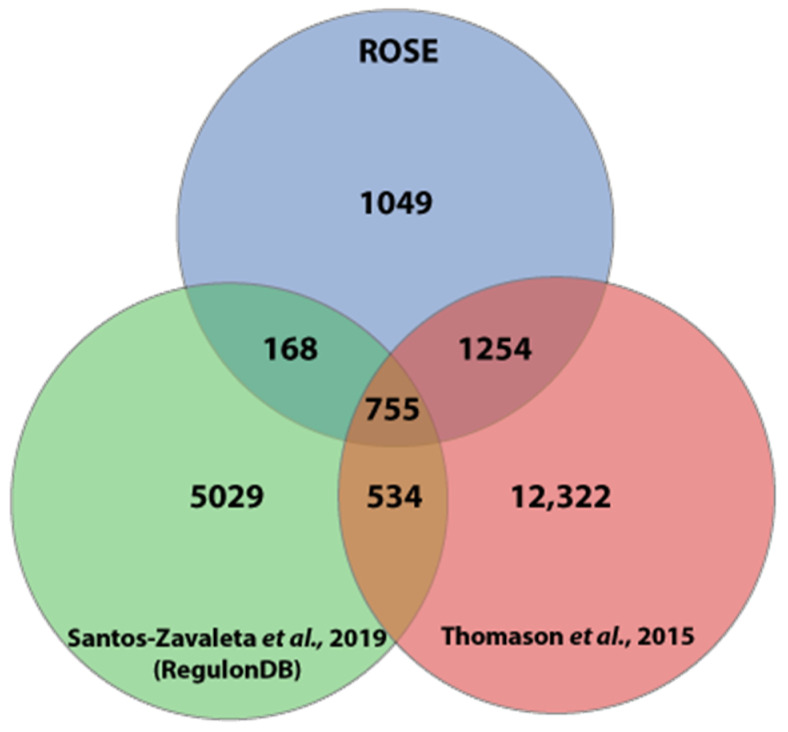
Comparison of transcription start sites (TSSs). TSS positions in the *E. coli* K-12 MG1655 genome from Thomason et al. [20], RegulonDB [22], and ROSE-Eσ70 (this study) have been compared. A difference of three nucleotides in either direction has been allowed.

**Figure 4 microorganisms-11-01388-f004:**
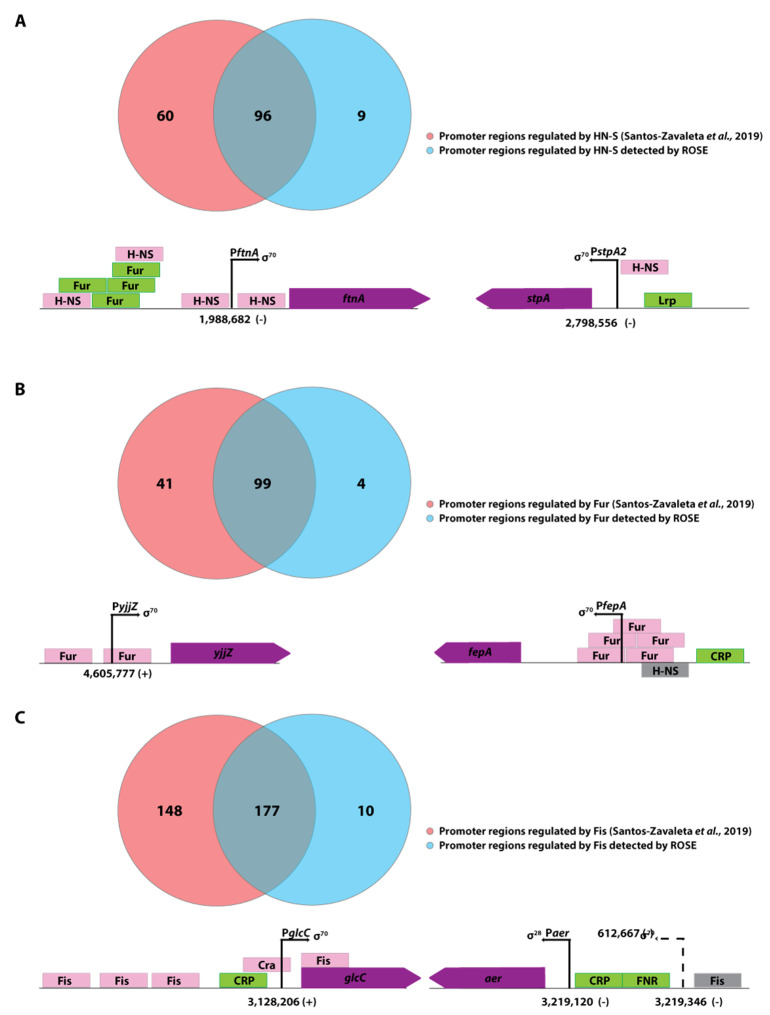
Analyses of the promoter regions regulated by the transcription factors H-NS, Fur and Fis with Venn diagrams and exemplary genomic regions. (**A**) H-NS with the illustration of promoter regions of *ftnA* and *stpA*, (**B**) Fur with the illustration of promoter regions of *yjjZ* and *fepA*, and (**C**) Fis with the illustration of promoter regions of *glcC* and *aer* described in RegulonDB [22] (red) and detected by ROSE-Eσ^70^ (blue). The genomic organization of the transcription units is depicted according to the EcoCyc database [27] (not to scale). The gene *aer* is illustrated with the known σ^28^-promoter *Paer* and the newly found TSS (dashed arrow) described by Thomason et al. [20] and identified by ROSE-Eσ^70^ and the Δ*fis* knockout strain.

**Figure 5 microorganisms-11-01388-f005:**
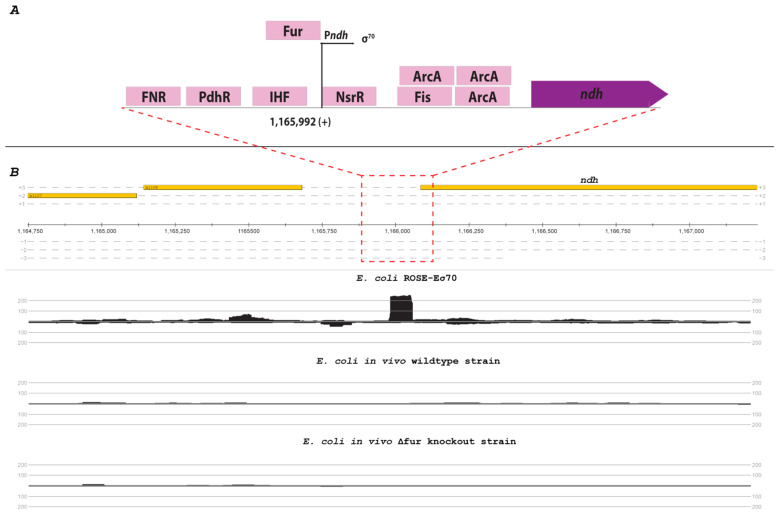
Promoter region and mapping results of the *ndh* gene. (**A**) Genomic organization of the transcription unit of *ndh* according to RegulonDB database [22] (not to scale). (**B**) Read count in the promoter region of *ndh* from the *E. coli* ROSE-Eσ^70^ (top), *E. coli* in vivo wildtype strain (middle), and *E. coli* in vivo Δ*fur* knockout strain. The mapping occurred on the respective reference genome (U00096.3) and is visualized with ReadXplorer [13].

## Data Availability

Coverage tracks imported into the UCSC genome browser session (only for access during reviewing period): https://genome.ucsc.edu/s/dbrandt/schmidt_brandt_K-12_MG1655 (accessed on 24 May 2023) The data discussed in this publication have been deposited in NCBI’s Gene Expression Omnibus [62] and are accessible through GEO Series accession number GSE159312 https://www.ncbi.nlm.nih.gov/geo/query/acc.cgi?acc=GSE159312 (accessed on 24 May 2023). To review GEO accession GSE159312: Go to https://www.ncbi.nlm.nih.gov/geo/query/acc.cgi?acc=GSE159312 (accessed on 24 May 2023) and enter token mtgpyqmullwlrcv into the box.

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
