# Peer review of "Characterization of Bacterial Transcriptional Regulatory Networks in Escherichia coli through Genome-Wide In Vitro Run-Off Transcription/RNA-seq (ROSE)"

_microorganisms, 2023, doi:10.3390/microorganisms11061388_

Round 1

Reviewer 1 Report

This is an interesting article, but there are some problems the author did not explain clearly, which need to be added in the subsequent revision.

Major issues:

1.         The author talks about characterizing the transcriptional regulatory network of bacteria, but I did not see it in the paper, please add more details. It is best to add a network diagram. This is very important.

2.         The difference between ROSE and RIViT-seq should be written in more detail.

3.    The tables in this paper should be changed to three lines.

4.    The format of the references needs to be carefully revised.

Other minor issues:

1.       Line 4 The abbreviation ROSE should be removed.

2.       Line 15 E. coli should be written in full when they first appear.

3.       Line 38 Escherichia coli should be abbreviated.

4.       Line 45 ROMA should be written in full when they first appear.

5.       Line 50 Bacillus subtilis should be italicized.

6.       Line 60 ROSE (Run-Off transcription/RNA-SEq) gaiwei Run-Off transcription/RNA-SEq (ROSE).

7.       Line 69 RIViT-seq should be written in full when they first appear.

8.       Line 59 Why is the data not shown?

9.       The resolution of the picture is not enough, so it needs to be modified and submitted again.

Minor editing of English language required

Author Response

Reviewer 1

This is an interesting article, but there are some problems the author did not explain clearly, which need to be added in the subsequent revision.
Major issues:
1.           The author talks about characterizing the transcriptional regulatory network of bacteria, but I did not see it in the paper, please add more details. It is best to add a network diagram. This is very important.

Answer: The method described in the paper is a tool for characterizing regulatory networks in bacteria and to complement the complex in vivo transcriptomic approaches by a simple in vitro approach. For validation of the method, in vivo experiments  were performed showing a broad overlap, but also interesting new results. Here, we did not intend to expand the regulatory networks of these transcription factors, since this would clearly require more experiments. Therefore, we refrained from presenting regulatory network maps. 
But we see the point of this reviewer that there is interesting information for researchers working on the repective regulatory networks.

Hence, we expanded Figure 4 by including Venn diagrams and the text of the Results section at lines 328-337 and the Discussion section at lines 505-509, detailing more on the results concerning novel genes potentially regulated by the three transcription factors. Also Supplementary Table 6 is new and comprises the list of novel TSS.

2.        The difference between ROSE and RIViT-seq should be written in more detail.

Answer: Not done. Our primary focus is on showcasing the strengths of our method, ROSE, rather than comparing it with other methods such as RIViT-seq. Such a comparison would also require that we carry out both methods in parallel, what we did not. For comparison, we have already provided a complete list of differences of these two methods in the Supplementary information (Suppl. Table S1). Additionally, we have already discussed the pros and cons of both methods in the Discussion section (lines 464-478). 
Considering these arguments, we feel unable to expand the manuscript as mentioned. However, if you feel that your request is not answered appropriately, please make concrete suggestions on where more detail is required.

3.       The tables in this paper should be changed to three lines.

Done. All tables are now changed into the three lines format.

4.        The format of the references needs to be carefully revised.

Done. The format of all references is changed into the format described in the microorganisms template of MDPI.

Other minor issues:
1.    Line 4 The abbreviation ROSE should be removed. 

Answer: Not done. We feel that the acronym should have a place in the title, also as a reminder to differentiate it from similar methods having the same princple approach, such as RIViT-seq. However, if you insist on this change, please inform.

2.    Line 15 E. coli should be written in full when they first appear. 
Done.

3.    Line 38 Escherichia coli should be abbreviated. 
Done.

4.    Line 45 ROMA should be written in full when they first appear. 
Done.

5.    Line 50 Bacillus subtilis should be italicized. 
Done.

6.    Line 60 ROSE (Run-Off transcription/RNA-SEq) gaiwei Run-Off transcription/RNA-SEq (ROSE). 
Done.

7.    Line 69 RIViT-seq should be written in full when they first appear. 
Done.

8.    Line 59 Why is the data not shown?  
During development of the method, the different buffer system was initially chosen due to a better compatibility with the downstream sequencing applications. In line with this, we noted that using the new buffer yielded a higher RNA yield. However, this increased yield was not relevant to the results presented in this study and our respective measurements were not analysed using statistical tools. 
We rewrote the sentence for clarity and removed the statement on the RNA yield (line 163).

9.    The resolution of the picture is not enough, so it needs to be modified and submitted again. 

Answer: We do not see which figure is meant. However, all figures except of the Venn diagram (Figure 3) are in vector format. The reason for a bad resolution is probably a bad conversion by the publisher’s web site. We will clarify this with the Editor’s office.

Comments on the Quality of English Language
Minor editing of English language required

Done. Some further typos were found and corrected. In addition, we utilized the computer program Grammarly and performed several small edits in the revised manuscript.

Reviewer 2 Report

The manuscript introduces a pioneering method, ROSE (Run-Off transcription/RNA-SEquencing), which provides a promising new avenue for the analysis of bacterial transcriptional networks. The method is remarkable in its simplicity, requiring only chromosomal DNA, ribonucleotides, the RNA polymerase (RNAP) core enzyme, and a specific sigma factor to recognize the corresponding promoters. This makes it both economical and practical for widespread use in various research settings.

While I have no major concerns, I recommend that the authors more explicitly outline the specific advantages and potential limitations of the ROSE method compared to existing techniques. Furthermore, the implications for future research and potential applications in various fields, such as microbiology, medicine, and biotechnology, should be discussed more thoroughly.

Author Response

While I have no major concerns, I recommend that the authors more explicitly outline the specific advantages and potential limitations of the ROSE method compared to existing techniques. Furthermore, the implications for future research and potential applications in various fields, such as microbiology, medicine, and biotechnology, should be discussed more thoroughly.

Dear Reviewer,

First of all, thank you for your kind words. We agree that the method is a useful tool to almost perfectly complement the classical methods for transcriptome analysis in vivo. I have two comments on our restraint to over-emphasize its importance in this manuscript. First is the existence of a similar method (RIViT-Seq), so we were not the first to publish this and second, we are currently writing two further manuscripts, one on extending the method by including transcription factors and small molecules on one hand and the other on extending it to other bacteria, by using homologous RNA polymerase preparations and a number of sigma factors.

Reviewer 3 Report

In this Article, the Authors invented, presented and applied a new method for genome-wide characterization of bacterial promoters called ROSE.  Using comparative transcriptome analysis, the Authors demonstrated that the method ROSE could identify bona fide promoters that were apparently repressed in vivo.  They claim that the new method ROSE is well-suited approach for characterizing transcriptional networks in bacteria.

This is a very complex and well done study that can be published in Microorganisms.

There are just two small concerns:

1 Abstract: It is quite difficult to read. Please, provide background and explain the importance of the developing a new method for bacterial promoter characterization.

2 Please, re- read several times the text to find typos and phrases with inappropriate style.

It would be good to re-read the text several times to correct phrases with inapreopriate style

Author Response

There are just two small concerns: 
1.     Abstract: It is quite difficult to read. Please, provide background and explain the importance of the developing a new method for bacterial promoter characterization. 
Done. In order to provide a background fort he development of the method, we introduced an opening statement into the abstract (lines 10-12).

2.     Please, re- read several times the text to find typos and phrases with inappropriate style. 
Done. Some further typos were found and corrected. In addition, we utilized the computer program Grammarly and performed several small edits in the revised manuscript.

Round 2

Reviewer 1 Report

The authors made careful revisions and agreed to accept the manuscript.

Minor editing of English language required.